# CLR-Bench: Evaluating Large Language Models in College-Level Reasoning

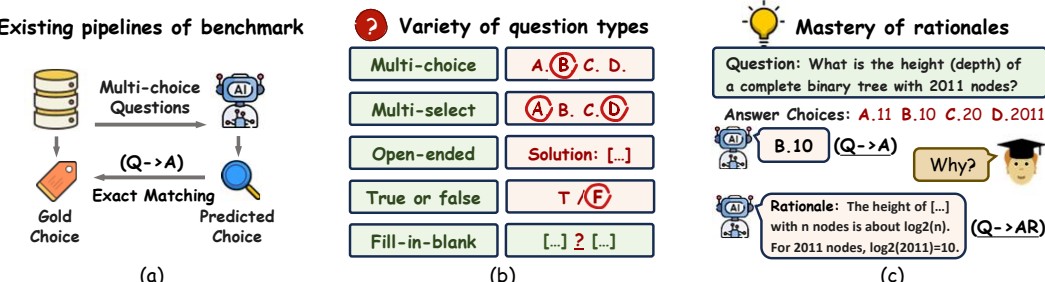

Figure 1: (a) illustrates a sketched overview of the current pipeline to benchmark LLMs with multi-choice questions, while our `CLR-Bench` proposes a more comprehensive one with a variety of question types and verifies the college-level reasoning ability of LLMs as shown in (b) and (c).

## Abstract

Large language models (LLMs) have demonstrated their remarkable performance across various language understanding tasks. While emerging benchmarks have been proposed to evaluate LLMs in various domains such as mathematics and computer science, they merely measure the accuracy in terms of the final prediction on multi-choice questions. However, it remains insufficient to verify the essential understanding of LLMs given a chosen choice. To fill this gap, we present `CLR-Bench` to comprehensively evaluate the LLMs in complex college-level reasoning. Specifically, $(i)$ we prioritize 16 challenging college disciplines in computer science and artificial intelligence. The dataset contains 5 types of questions, while each question is associated with detailed explanations from experts. $(ii)$ To quantify a fair evaluation of LLMs' reasoning ability, we formalize the criteria with two novel metrics. Q→A is utilized to measure the performance of direct **a**nswer prediction, and Q→AR effectively considers the joint ability to **a**nswer the question and provide **r**ationale simultaneously. Extensive experiments are conducted with 40 LLMs over 1,018 discipline-specific questions. The results demonstrate the key insights that LLMs, even the best closed-source LLM, i.e., GPT-4 turbo, tend to '***guess***' the college-level answers. It shows a dramatic decrease in accuracy from 63.31% Q→A to 39.00% Q→AR, indicating an unsatisfactory reasoning ability.

## 1 Introduction

The emerging large Language Models (LLMs), e.g., GPT-4 (Achiam et al., 2023), Gemini (Team et al., 2023) and Llama (Touvron et al., 2023b) have performed significantly across various natural language understanding tasks (Hendrycks et al., 2020). They are tightly integrated into the downstream scenarios, such as college education and medicine (Meyer et al., 2023; Abd-Alrazaq et al., 2023), thereby transforming the paradigm for both research and applications. In response to this advancement, several Off-the-shelf benchmarks have been developed to assess LLMs across different domain-specific tasks. For instance, MMLU (Hendrycks et al., 2020), as the most influential benchmark, examines LLMs across 57 subjects in STEM. GSB8K (Cobbe et al., 2021) and MATH (Hendrycks et al., 2021) particularly focus on model performance over mathematics problems. CMMLU (Li et al., 2023) and C-Eval (Huang et al., 2024), designed for testing LLMs over Chinese language understanding tasks. While these benchmarks are valuable for measuring the question answering capabilities of LLMs, they

still fall short of evaluating the depth of reasoning that reflects true understanding subject to different topics. Followed by MMLU-Pro (Wang et al., 2024b), they have realized this situation and proposed to enlarge the dataset with more reasoning-focused questions and expand the choices from four to ten.

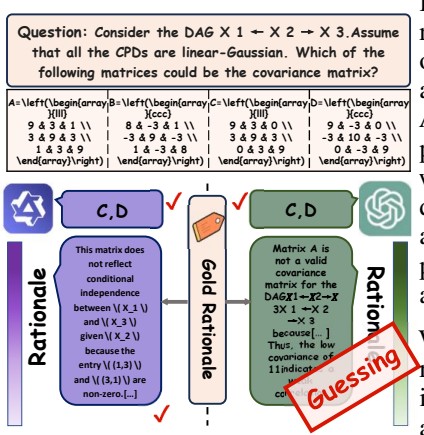

However, it is still unsatisfying since their evaluations also rely on multi-choice questions, measuring performance based on the correctness of the final prediction. We would like to ask: '***do LLMs really know the rationale or just guess?***' An example is illustrated in Figure 2, the current paradigm potentially enables LLMs to rely on the choices for inference without genuinely understanding the rationale behind the question. Moreover, the underlying rationale behind the answer remains unverified. Assessment of the reasoning process becomes a critical limitation of existing benchmarks and is urged to be carefully designed for social good.

Figure 2: A real question answered by Qwen 2.5 and GPT-3.5. While GPT-3.5 has successfully identified the correct choices, it made the wrong rationale indicating the potential 'guessing' behavior.

We are thereby motivated to comprehensively evaluate the reasoning capability of LLMs through rationale. However, it still remains challenging for three practical reasons. First, a high-quality multi-type question-answering dataset is not readily available for reasoning tasks. It remains unexplored for a systematic way to collect questions across different types and topics while multi-choice questions are dominating existing pipelines. Second, annotating rationales for each question is laborious. It requires much effort from domain experts to indicate the necessary knowledge points or inference procedures that should be included in the rationale. Third, there are no standard criteria for evaluating the generated rationales and performance of rationalizing. How to determine an authentic level of reasoning ability remains a tough task that requires expensive and manual efforts.

To bridge the gap, we present `CLR-Bench`, currently the most challenging and comprehensive benchmark particularly designed to evaluate LLMs in college-level reasoning. Our benchmark goes beyond simple prediction accuracy by focusing on the underlying reasoning processes that lead to those answers. To be specific, we prioritize 16 authoritative college textbooks in computer science and artificial intelligence for high-quality reasoning questions. ($i$) We first involve domain experts to construct a hierarchical topic graph (HTG), where the condensed topics are categorized into three levels. For instance, *programming_fundamentals*, as a level-1 topic, owns ten level-2 topics as child nodes such as *Variable*, *Operator*, *Pointer and reference*, etc. The HTG is utilized to effectively guide the question collection per topic. In total, `CLR-Bench` features a dataset comprising 1,018 discipline-specific questions, in the form of five distinct question types: *multiple-choice* (MC), *multiple-select* (MS), *true/false* (TF), *open-ended* (OE), and *fill-in-the-blank* (FB). ($ii$) We carefully design an expert-guided prompting strategy to facilitate the gold rationale generation by GPT-4o. Each type of question is expected to have different content for explanations. For example, the rationale for MC and MS questions is required to include the reason for choosing the predicted choice as well as the reasons for not choosing the remaining ones. While for OE questions, the rationale should include the relevant information and necessary knowledge points involved to give the solutions. ($iii$) We introduce two novel metrics, i.e., Q→A and Q→AR to tackle the challenge of measuring the genuine performance of reasoning ability. Specifically, LLMs are expected to predict the answer (A) and provide the corresponding rationale (R) at the same time. A standardized criterion is tailored for Q→AR based on the accuracy score of Q→A and Q→R. Through our proposed `CLR-Bench`, we obtain several insightful observations based on the results and summarize our contributions below:

**Contributions and findings:**

▶ We release a carefully constructed multi-type rationalized dataset, as currently the most challenging reasoning task at the college level, comprising 1,018 questions in 5 types spanning 16 subjects in computer science and artificial intelligence. We first formalize the paradigm of multi-type dataset construction under the guide of the hierarchical topic graph. It lays a rigorous foundation for reasoning-based research and evaluation.

▶ We introduce two novel evaluation metrics: Q→A and Q→AR. They evaluate both the answer and the rationale provided, thereby capturing a more comprehensive view of LLMs' reasoning abilities. We also instantiate the detailed criteria for grading Q→AR based on Q→A and Q→R. This leads to exciting observations indicating that ($i$) LLMs tend to '**guess**' answers for college-level questions,

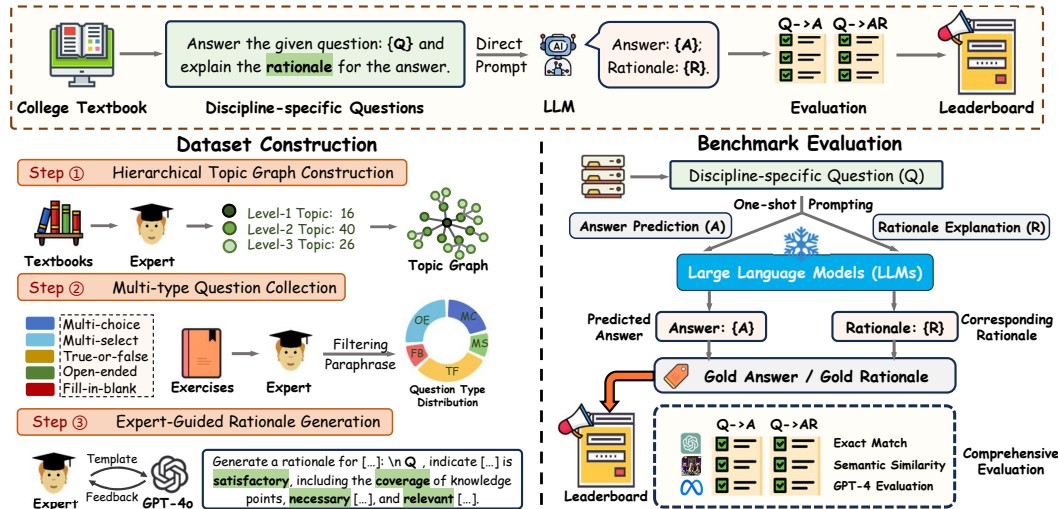

Figure 3: The overview of our proposed CLR-Bench. **Dataset Construction**. Domain experts first curate a condensed hierarchical topic graph to guide the collection of five types of questions. GPT-4o is then carefully instructed to assist the experts in gold rationale generation. **Benchmark Evaluation**. We formally define standardized criteria for each type of question and the corresponding rationale. .

instead of truly understanding the rationale with a dramatically dropping Q→AR. (*ii*) Model sizes do **not** inherently guarantee superior performance in Q→AR, despite larger models often achieving higher accuracy in Q→A. Several smaller models notably exhibit stronger performance in Q→AR, surpassing larger ones to provide accurate and coherent rationales.

▶ We conduct extensive experiments involving 40 LLMs including both open- and close-source ones. Sufficient analysis has been provided based on the results and observations across different dimensions, i.e., question types, topics, and model families.

## 2 TASK STATEMENT

In order to systematically evaluate the reasoning capabilities of LLMs, CLR-Bench incorporates a diverse set of question types that reflect various challenges. Each question type is carefully designed to test different aspects to verify LLM's mastery of in-depth rationalizing. Moreover, given a question, LLMs are required to provide rationales to prove their understanding beyond the direct prediction. In this section, we formally define the question types and the rationale we expect for each type.

### 2.1 MULTI-TYPE REASONING TASKS

**Multi-choice** (MC) questions present a single question with multiple possible answers including disturbing choices, from which LLMs must select the most appropriate one. This type of question evaluates an LLM's ability to discern and provide the correct answer from a set of plausible options; **Multi-select** (MS) questions are similar to multi-choice questions but require LLMs to select more than one correct answer. The more disturbing situations assess the LLM's capability to handle more complex queries where multiple factors must be considered. Moreover, LLMs also face the dilemma of either choosing more for the full mark or selecting the most confident one to get half of the score; **Fill-in-blank** (FB) questions require LLMs to complete a sentence or statement with the appropriate term or phrase, measuring how well the LLM could generate contextually relevant content; **Open-ended** (OE) questions allow for a wide range of responses, requiring LLMs to formulate detailed and comprehensive answers. It tests the LLM's ability to generate well-rounded explanations; **True-or-false** (TF) questions require students to determine the veracity of a given statement. This type examines the LLM's ability to assess factual accuracy based on their knowledge.

### 2.2 FORMALIZATION OF RATIONALE

In this subsection, we define the expected rationale for each question type in CLR-Bench. The rationale must demonstrate not just correctness but a clear understanding of the contexts.

Table 1: Comparisons among existing benchmarks and our proposed `CLR-Bench`.

| Benchmark | Question Types | Rationale | Evaluation Metric |
|---|---|---|---|
| MMLU (Hendrycks et al., 2020) | MC | ✗ | Q→A |
| MMLU-Pro (Wang et al., 2024b) | MC | ✗ | Q→A |
| CMMLU (Li et al., 2023) | MC | ✗ | Q→A |
| C-Eval (Huang et al., 2024) | MC | ✗ | Q→A |
| GSB-8K (Cobbe et al., 2021) | MC | ✗ | Q→A |
| MATH (Hendrycks et al., 2021) | MC | ✗ | Q→A |
| `CLR-Bench` (Ours) | MC, MS, TF, OE, FB | ✔ | Q→A, Q→AR |

**Interference Recognition for MC**: The rationale should explain why the selected option is correct and why the disturbing alternatives are incorrect, demonstrating clear reasoning and a solid grasp of the underlying principles while eliminating the interference impacts from incorrect choices;

**Complexity Management for MS**: For multi-select questions, the rationale is expected to justify each correct selection and explain why the other options are wrong, not only reflecting the LLM's knowledge reserve and an anti-disturbance ability, but also decision-making performance when facing the dilemma of more marks but higher risks, or a safer selection but lower accuracy;

**Mastery of concepts for TF**: The rationale should concisely state the concepts behind the truth value, citing relevant facts or rules that support the decision;

**Conceptual understanding for FB**: Explaining FB questions requires reasons why the chosen word or phrase fits the sentence, showing an understanding of context and meaning, as well as the underlying knowledge or concepts in the statement;

**Open reasoning ability for OE**: As the most difficult task, the rationale for OE, different from the corresponding solution, should provide a structured, detailed, and well-rounded explanation, covering all relevant aspects and reflecting comprehensive reasoning steps to infer the final answer.

By formalizing the rationale for each type of question, `CLR-Bench` ensures that LLMs are evaluated not only on their ability to provide correct answers but also on their capability to reason effectively and justify their choices. A sketched comparison is shown in Table 2.2, indicating that our benchmark significantly moves beyond surface-level correctness in existing benchmarks to assess the depth of understanding, logical coherence, and mastery of the subject matter.

## 3 CONSTRUCTION DETAILS OF CLR-BENCH

To ensure the quality of our benchmark, We recruit eight domain experts to handle 16 college-level disciplines. Their expertise is maximally leveraged through our novel three-stage strategy. In this section, we introduce the core procedures of constructing a high-quality dataset leveraging both expert knowledge and GPT-4o as a tool.

### 3.1 HIERARCHICAL TOPIC GRAPH

A hierarchical topic graph (HTG) $\mathcal{G}$ is effectively designed to support multiple disciplines, ensuring that key topics across computer science and artificial intelligence are captured in a way that mirrors their complexity and interrelationships. To comprehensively model discipline-specific knowledge, we refer to the table-of-content in the textbooks where the multi-level structure ensures that the graph reflects high-level overviews and detailed subtopics to guide the question collection process.

**DEFINITION:** Hierarchical Topic Graph. Let $\mathcal{O}$ be a well-structured ontology for constructing a HTG $\mathcal{G} = \{\mathcal{O}, \mathcal{E}, \mathcal{R}\}$, where $\mathcal{O} = \{\mathcal{T}, \mathcal{R}\}$ defines the topics $\mathcal{T}$ and relations $\hat{\mathcal{R}}$ between domain-specific topics $\mathcal{E}$. The ontology $\mathcal{O}$ is derived from expert knowledge in 16 disciplines and structured around a multi-level hierarchy. Specifically, $\mathcal{T}$ consists of three levels of topics, where each level represents increasingly specific subtopics.

Each triple $(h, r, t)$ in $\mathcal{G}$ comprises a head topic $h$, a relation $r$, and a tail subtopic $t$, where $h, t \in \mathcal{T}$ and $r \in \mathcal{R}$. We define the relations $\mathcal{R}$ as "has_subtopic" to represent the hierarchical structure, capturing the relationship between general topics and their more granular subtopics. For instance, in

Table 2: `CLR-Bench` leaderboard of various models on the Q→AR task. Each model is expected to answer correctly and rationalize accurately at the same time to get 1 point for each question.

| # | Models | Q→A (Direct performance of prediction) | | | | | | Q→AR (Reasoning performance of rationale) | | | | | |
|---|--------|-----|-----|-----|-----|-----|-----|-----|-----|-----|-----|-----|-----|
| | | MC | MS | TF | FB | OE | Q→A | MC | MS | TF | FB | OE | Q→AR |
| 1 | qwen2.5-7b | 69.12% | 54.95% | 54.11% | 40.00% | 49.07% | 54.62% | 43.55% | 34.46% | 43.43% | 45.00% | 8.36% | 33.37% |
| 2 | gemma2-9b-it | 72.81% | 50.45% | 55.70% | 43.81% | 40.52% | 53.54% | 45.62% | 32.88% | 47.15% | 44.52% | 7.90% | 34.63% |
| 3 | qwen2.5-72b | 80.18% | 73.42% | 60.44% | **47.62%** | 52.04% | 62.52% | 50.00% | 43.02% | 45.97% | **49.52%** | 11.99% | 37.89% |
| 4 | gemma2-9b | 16.13% | 47.75% | 11.71% | 9.52% | 11.34% | 16.26% | 11.29% | 8.33% | 10.60% | 11.67% | 3.62% | 8.77% |
| 5 | mixtral-8x7b-instruct-v0.1 | 59.91% | 45.05% | 34.18% | 16.19% | 43.12% | 41.36% | 42.97% | 31.31% | 39.08% | 34.52% | 8.36% | 30.48% |
| 6 | phi-3-medium-4k-instruct | 74.65% | 63.96% | 56.01% | 38.10% | 48.88% | 57.12% | 51.50% | 33.56% | 49.68% | 44.05% | 11.34% | 37.60% |
| 7 | qwen2.5-72b-instruct | **81.11%** | **77.93%** | **63.61%** | 45.71% | **52.23%** | **64.05%** | 50.58% | **45.50%** | 53.56% | 48.10% | **13.10%** | 40.79% |
| 8 | llama-2-7b | 54.38% | 15.77% | 44.62% | 19.05% | 36.43% | 38.75% | 26.61% | 18.47% | 27.29% | 24.05% | 6.78% | 20.43% |
| 9 | llama-3.1-8b-instruct | 63.13% | 51.80% | 56.96% | 25.71% | 41.82% | 50.49% | 39.06% | 22.97% | 41.85% | 33.57% | 8.55% | 29.54% |
| 10 | phi-3-mini-4k-instruct | 69.12% | 59.46% | 55.38% | 34.29% | 44.80% | 53.78% | 49.42% | 37.84% | 45.41% | 41.43% | 9.57% | 35.56% |
| 11 | yi-1.5-6b-chat | 37.33% | 53.15% | 48.10% | 25.71% | 38.85% | 41.60% | 30.30% | 32.43% | 32.99% | 34.05% | 6.88% | 25.56% |
| 12 | deepseek-7b-chat | 49.31% | 27.48% | 45.57% | 20.00% | 31.41% | 38.02% | 30.76% | 18.92% | 34.97% | 27.62% | 5.11% | 23.67% |
| 13 | yi-1.5-34b-chat | 68.66% | 53.60% | 56.96% | 40.95% | 39.96% | 52.95% | 40.90% | 33.33% | 48.34% | 44.76% | 7.16% | 33.87% |
| 14 | deepseek-7b-base | 49.77% | 36.49% | 44.94% | 18.10% | 36.62% | 40.08% | 27.88% | 18.47% | 28.16% | 25.24% | 5.39% | 20.73% |
| 15 | llama-3.1-70b-instruct | 80.18% | 74.77% | 61.39% | 40.00% | 44.61% | 60.22% | 44.24% | 38.06% | 45.09% | 38.57% | 7.99% | 33.67% |
| 16 | llama-3-8b | 59.91% | 35.14% | 50.95% | 26.67% | 43.31% | 46.61% | 32.37% | 24.10% | 31.09% | 29.29% | 7.53% | 24.19% |
| 17 | mistral-7b-instruct-v0.1 | 57.60% | 45.05% | 40.82% | 21.90% | 42.19% | 43.27% | 35.48% | 27.48% | 34.41% | 33.33% | 6.04% | 26.28% |
| 18 | gemma-7b-it | 15.21% | 37.84% | 45.25% | 8.57% | 13.38% | 25.83% | 13.02% | 1.80% | 35.52% | 20.24% | 10.04% | 18.74% |
| 19 | qwen1.5-7b-chat | 32.26% | 50.00% | 11.71% | 13.33% | 35.32% | 26.67% | 27.42% | 26.58% | 27.22% | 33.57% | 6.41% | 22.35% |
| 20 | yi-1.5-6b | 53.00% | 59.01% | 51.27% | 30.48% | 42.75% | 48.08% | 33.53% | 29.95% | 39.16% | 35.00% | 6.13% | 27.80% |
| 21 | qwen2.5-32b-instruct | 80.18% | 75.23% | 63.29% | 44.76% | 52.04% | 63.31% | **55.41%** | **45.50%** | **56.09%** | 48.33% | 11.80% | **42.29%** |
| 22 | qwen2.5-7b-instruct | 70.97% | 63.06% | 58.23% | 37.14% | 49.26% | 56.93% | 52.65% | 34.23% | 50.32% | 41.90% | 8.92% | 37.25% |
| 23 | llama-3.1-8b | 62.67% | 40.54% | 51.58% | 30.48% | 44.98% | 48.82% | 34.22% | 28.83% | 32.52% | 33.57% | 7.99% | 26.11% |
| 24 | llama-3.2-3b-instruct | 57.14% | 39.19% | 49.05% | 23.81% | 34.94% | 43.37% | 40.55% | 21.40% | 38.29% | 32.62% | 8.09% | 28.36% |
| 25 | llama-3-8b-instruct | 60.83% | 49.55% | 52.22% | 29.52% | 47.40% | 50.15% | 43.78% | 26.35% | 41.30% | 36.19% | 9.76% | 31.34% |
| 26 | openchat-3.5 | 60.83% | 39.64% | 49.68% | 25.71% | 36.25% | 44.94% | 35.94% | 22.75% | 33.86% | 30.24% | 8.46% | 26.01% |
| 27 | qwen1.5-7b | 58.99% | 54.50% | 47.15% | 37.14% | 38.29% | 47.10% | 36.64% | 33.11% | 32.99% | 35.95% | 6.78% | 27.16% |
| 28 | llama-2-7b-chat | 38.25% | 36.49% | 39.24% | 13.33% | 35.50% | 35.07% | 26.04% | 16.89% | 31.49% | 26.67% | 5.30% | 21.32% |
| 29 | mistral-7b-v0.1 | 58.06% | 42.79% | 50.63% | 37.14% | 43.87% | 48.18% | 33.06% | 29.05% | 34.10% | 35.24% | 7.16% | 26.33% |
| 30 | yi-1.5-34b | 70.97% | 58.11% | 57.59% | 41.90% | 48.33% | 56.43% | 40.55% | 35.59% | 42.96% | 43.10% | 7.25% | 32.22% |
| 31 | mixtral-8x7b-v0.1 | 67.28% | 39.64% | 53.48% | 39.05% | 45.72% | 51.38% | 39.06% | 31.53% | 36.00% | 36.19% | 8.83% | 29.00% |
| 32 | llama-3.1-70b | 75.58% | 51.80% | 57.59% | 45.71% | 47.77% | 56.97% | 43.89% | 35.81% | 41.46% | 45.48% | 10.50% | 33.60% |
| 33 | llama-3.2-1b-instruct | 41.47% | 40.99% | 37.66% | 14.29% | 25.09% | 33.10% | 28.00% | 13.06% | 24.68% | 24.76% | 6.69% | 19.38% |
| 34 | gpt-3.5-turbo | 63.13% | 66.67% | 58.54% | 23.81% | 47.77% | 53.98% | 42.51% | 41.22% | 49.60% | 39.29% | 6.41% | 34.70% |
| 35 | claude-3-sonnet | 76.96% | 76.58% | 59.81% | 42.86% | 48.33% | 60.51% | 50.69% | 44.59% | 48.26% | 45.95% | 11.80% | 38.51% |
| 36 | gemini-1.5-pro | 78.34% | 71.62% | 55.38% | 46.67% | **52.42%** | 60.36% | 50.46% | 40.54% | 43.35% | 50.24% | 12.83% | 37.21% |
| 37 | deepseek-chat | 78.80% | 77.03% | 62.66% | 40.95% | 51.30% | 62.43% | **55.41%** | 44.14% | 55.14% | 46.43% | **13.48%** | 42.09% |
| 38 | gpt-4o | **84.33%** | 76.58% | **63.92%** | **54.29%** | 34.01% | 60.76% | 53.11% | 44.82% | **57.52%** | 51.19% | 8.55% | 41.60% |
| 49 | gpt-4-turbo | 82.49% | 78.38% | **63.92%** | 50.48% | 45.91% | 63.31% | 51.73% | **45.95%** | 49.45% | **51.43%** | 8.74% | 39.00% |
| 40 | claude-3-opus | 80.18% | **81.53%** | 62.66% | 52.38% | 44.42% | 62.57% | 50.35% | 43.02% | 50.08% | 50.00% | 9.20% | 38.56% |

the domain of computer science, let $h$ = "Programming fundamentals", $r$ = "*has_subtopic*", and $t$ = "Variable". Further, we define the next level of granularity with the triple (Variable, *has_subtopic*, Constant). The construction of $\mathcal{G}$ begins with a careful review of tables of contents and chapter titles from relevant textbooks or curriculum guides. Experts from each discipline assist in condensing the most critical topics and ensuring that the hierarchical relationships are accurate and representative of the disciplines's knowledge structure.

## 3.2 MULTI-TYPE QUESTION COLLECTION

Given the well-structured HTG, a rigorous process is employed to collect questions spanning a wide range of topics in computer science and artificial intelligence. Experts manually curate the question set based on a structured ontology, which organizes knowledge into three levels: 16 level-1 topics, 40 level-2 topics, and 26 level-3 topics. Each level reflects a progression from broad subject areas to more specific subtopics. For each concept within these $n$ topics, experts are required to have at least two questions of each question type, i.e., $2 \times 5 \times n$, to maintain a balanced and comprehensive coverage of the content. To ensure high quality, experts refer to authoritative sources such as textbooks, course materials, and widely accepted curricula. Each question is then reviewed to verify its relevance, clarity, and alignment with its respective concept. The process ensures the dataset reflects real-world educational standards while encompassing the full breadth and depth of the domain.

## 3.3 EXPERT-GUIDED RATIONALE GENERATION

One of the key contributions of `CLR-Bench` is the incorporation of expert-verified rationales, which test whether LLMs truly understand the underlying concepts behind their answers. While manually generating detailed rationales for thousands of questions would be both time-consuming and resource-intensive, we streamline this process by integrating GPT-4o alongside expert knowledge. This hybrid approach maximizes both efficiency and quality, reducing the manual burden on experts while ensuring that the rationales produced are both comprehensive and accurate. The detailed prompt templates for expert-guided rationale generation are illustrated in Appendix C.

The process operates in a feedback loop where experts provide carefully designed prompts to GPT-4o, which generates initial rationale drafts. These drafts include all necessary explanations, logical reasoning, and relevant information tied to the question. Experts then review the output, making refinements or modifications as needed to meet high-quality standards. This collaborative approach leverages the speed and coverage of GPT-4o while maintaining the nuanced accuracy that expert verification ensures. By combining human expertise with the capabilities of GPT-4o, we can produce high-quality rationales at a fraction of the cost and time of fully manual methods.

By using GPT-4o as an initial rationale generator, we significantly reduce the effort required for manual rationale creation, making the process scalable across large datasets. Experts no longer need to write rationales from scratch but can instead focus on refining and verifying the generated explanations. This hybrid model ensures both speed and quality in rationale production, enabling us to maintain high standards while accommodating the breadth of questions across multiple disciplines. Rather than assessing LLMs purely on their ability to select correct answers, we evaluate their capacity to generate well-structured, coherent explanations that demonstrate true comprehension. This shift from answer prediction to reasoning evaluation is fundamental in advancing LLMs for real-world applications, where understanding and articulation are critical. In general, `CLR-Bench` comprises 1,018 college-level discipline-specific questions. Among them are 217 for MC task, 111 for MS, 316 for TF, 269 for OE, and 105 for FB, in a relatively balanced distribution.

## 4 EVALUATION AND EXPERIMENTAL ANALYSIS

In this section, we formally define our novel evaluation protocol with two new metrics, especially introducing the criteria on reasoning performance in terms of Q→AR. Insightful analysis is provided through systematically evaluating 40 state-of-the-art LLMs spanning both open and closed-source over 1,018 questions in `CLR-Bench`.

In this part, we would like to highlight one of our contributions by introducing a novel evaluation protocol considering the performance of both final prediction (Q→A) and rationale (Q→AR). We provide a detailed demonstration of a fair measurement for each type of question.

**Q→A**. ($i$) Exact Matching is applied when the expected answer is discrete and well-defined, as in MC, MS, TF, and FB questions. ($ii$) Semantic Similarity. The evaluation of OE questions remains a hard problem for the community. We leverage RoberTa-large (Liu et al., 2019) to sketchily compute the semantic similarity between the gold solution and the provided version by LLM. The threshold is set at 0.90, which means that the solution with higher scores will be treated as correct, while the remaining questions will be passed to the next step. ($iii$) GPT-4o-assisted Expert Evaluation. To improve the experts' efficiency and maximize their efforts on OE evaluation, we have carefully designed a prompt template that asks GPT-4o to provide a draft including the essential reasoning steps and relevant information. Experts will then verify the draft and fix the final gold rationale one by one to ensure accuracy and completeness.

Before evaluating Q→AR that measures the ability to answer and rationalize at the same time, we introduce another intermediate metric for rationale only, i.e., **Q→R**. ($i$) Semantic Similarity. We compute the similarity score between the LLM-generated rationale and a gold-standard rationale provided by experts, filtering out rationales that are semantically close to the label. ($ii$) GPT-4-assisted Expert Evaluation. Similarly, we process the rationales that are marked lower than 0.9 with the help of GPT-4o and experts. Differently, we have a set of particularly designed templates for each question type, details of all the prompts used for Q→A and Q→R are introduced in Appendix D. All the rationales will be marked $\{0, 0.5, 1\}$ for the score.

**Q→AR**. Based on the previously computed Q→A and Q→R score, we could automatically calculate the Q→AR score accordingly. We enforce a strict evaluation to punish the potential 'guessing' behavior, particularly in cases where the model provides a correct answer but is associated with an incorrect or incomplete rationale. Specifically, ($i$) if both the answer and the rationale are correct, the model is awarded a full score of 1.0. ($ii$) However, if the answer is correct but the rationale is only partially correct, the score is reduced to 0.5. ($iii$) When the answer is correct but the rationale is completely wrong, the score is significantly penalized, dropping to 0.0. ($iv$) On the other hand, if the answer is incorrect but the rationale is correct, the model still receives some credit with a score of 0.5, and with a partially correct rationale, the score becomes 0.25. ($v$) If both the answer and rationale are

Table 3: Evaluation Methods for Each Question Type under Q→A and Q→AR.

| Question Type | Q→A Evaluation | Q→AR Evaluation |
|---|---|---|
| MC | Exact Matching for choice $c$ | Semantic Similarity
Expert Evaluation |
| MS | 0 for incorrect $\hat{c}$
0.5/1 for partial $\{c\}/\{c\}$ | Semantic Similarity
Expert Evaluation |
| TF | Exact Matching for T/F | Semantic Similarity
Expert Evaluation |
| FB | Exact Matching | Semantic Similarity
Expert Evaluation |
| OE | Semantic Similarity
Expert Evaluation | Semantic Similarity
Expert Evaluation |

incorrect, the score remains at 0.0. This ensures that models are not unfairly rewarded for guessing correct answers without demonstrating understanding through their rationales.

### 4.1 ONE-SHOT SETTING

Unlike benchmarks such as MMLU (Hendrycks et al., 2020) that evaluate LLMs in five-shot, we introduce a standardized one-shot setting in `CLR-Bench` to ensure uniformity across models, particularly for smaller LLMs. Experts first maintain a fixed set with a single example per question type and discipline. The example guides the LLMs in structuring their responses while preventing them from leveraging extensive context. For instance, LLMs are required to output in the format of `{Rationale:{}, Answer:{}}`. This constraint ensures a consistent output format across all models, promoting fairness in evaluation, especially when comparing large and small LLMs. By using one-shot examples, we also aim to highlight the fairness in `CLR-Bench` of evaluating the reasoning capabilities without potential over-reliance on multiple context examples.

### 4.2 LEADERBOARD

For benchmarking widely used large language models (LLMs), we selected around 40 models from various model families and series. This includes popular open-source LLM families such as LLaMA (Touvron et al., 2023a;b; Dubey et al., 2024), Qwen (Bai et al., 2023; Yang et al., 2024), Phi (Abdin et al., 2024), and Mistral (Jiang et al., 2023; 2024), as well as powerful proprietary models like GPT (Brown, 2020; Achiam et al., 2023) and Claude (Anthropic, 2023a).

LLaMA (Dubey et al., 2024) is currently one of the most prominent LLM architectures, with model sizes ranging from 1B to 405B parameters across five generations. Phi (Abdin et al., 2024) and Gemma (Team et al., 2024a;b) families extend context length while maintaining relatively smaller model sizes, achieving performance comparable to larger models. Qwen (Yang et al., 2024), Yi (Young et al., 2024), and DeepSeek (Bi et al., 2024; Liu et al., 2024) stand out as LLMs excelling in areas such as mathematics, coding, and language understanding, benefiting from specialized downstream fine-tuning (Hui et al., 2024). The Mistral (Jiang et al., 2023) family, leveraging a Mixture of Experts (MoE) architecture (Jiang et al., 2024), excels in reasoning while maintaining high efficiency.

Among proprietary LLMs, the GPT series, widely recognized through ChatGPT (Ray, 2023) and GPT-4 (Achiam et al., 2023), is regarded as one of the most successful and powerful in the AI community. Extensive training data and well-crafted architectures enable these models to handle highly complex and challenging tasks. The Claude (Anthropic, 2023a) series is particularly adept at understanding nuanced and complex human instructions, while the Gemini (Team et al., 2023) series demonstrates strong capabilities in multi-language understanding and translation, excelling in long-context retrieval and inference.

### 4.3 OBSERVATIONS AND DISCUSSIONS

The results in the `CLR-Bench` leaderboard highlight notable discrepancies between the models' ability to answer questions correctly (Q→A) and their capacity to rationalize those answers accurately (Q→AR). In this section, we highlight four insightful observations and conduct corresponding discussions hereunder.

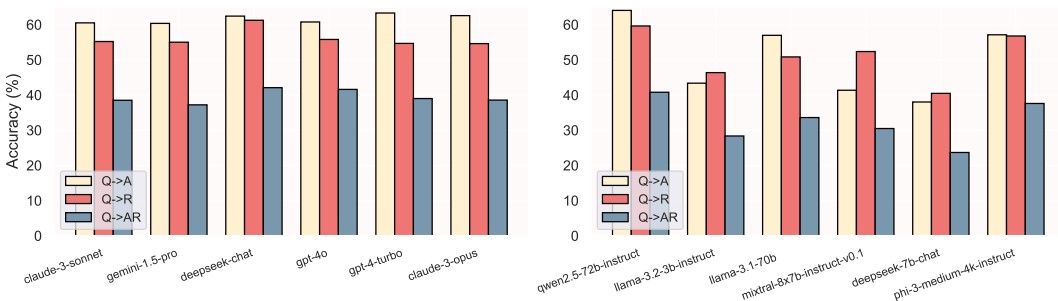

Figure 5: We selectively show the performance among closed- and open-source LLMs as well as within each group of models. The comparisons are made based on Q→A, Q→R and Q→AR.

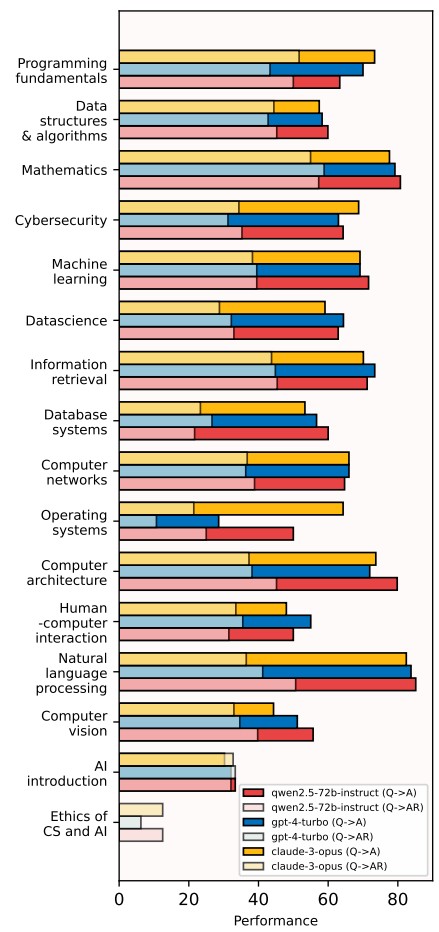

Figure 4: A discipline-level model performance in terms of both Q→A and Q→AR. We selectively choose Qwen 2.5 72b, GPT-4 turbo and Claude3-opus to showcase their mastery of the reasoning ability across different topics.

We summarize the main performance in Table 2. Subject to Q→A, Qwen2.5-72b-Instruct, GPT-4-turbo, and Qwen2.5-72b consistently outperform other models with 64.05%, 63.31%, and 62.52% respectively, showing their strong comprehension ability in providing correct answers across tasks. Other large-scale models like Llama-3.1-70b-Instruct and Phi-3-Medium-4K-Instruct also demonstrate relatively high performance, achieving 57.12% and 56.97% in the Q→A category. While smaller models such as Gemma2-9b and Gemma-7b-IT struggle significantly, with Q→A scores of 16.26% and 25.83% respectively. In general, while some top-performing models show promise, the overall performance of LLMs—including both closed- and open-source models remains relatively unsatisfactory. It also reflects the high quality and complexity of our dataset, emphasizing the need for further advancements in college-level question answering.

**LLMs tend to 'guess' answers**. A detailed analysis of Q→AR scores reveals that despite their relatively high accuracy in answering questions (Q→A), many LLMs struggle significantly to provide coherent and accurate rationales. For instance in Figure 5, while Qwen2.5-72b-Instruct leads the leaderboard in Q→A with a score of 64.05%, the corresponding Q→AR score drops dramatically to 40.79%. This discrepancy suggests that LLMs are probably 'guessing' the correct answers without a deep understanding of the underlying reasoning. This phenomenon becomes more evident in models such as GPT-4-turbo, which achieves a strong Q→A performance (63.31%) but also exhibits a noticeable drop to 39.52% in Q→AR. We have provided sufficient examples of this potential guessing behavior in the Appendix E.

Such a disparity indicates that LLMs may rely on superficial pattern recognition or shortcuts to arrive at correct answers, without truly grasping the logical foundations or the underlying knowledge required to justify their decisions. This raises concerns about the robustness of these models in educational settings, where the ability to reason and explain is just as important as providing the correct answer.

**LLMs are not good at non-MC questions**. We visualize the radar diagrams to showcase the expertise of LLMs on different types of questions in Figure 6. Performance on non-MC questions presents another significant challenge for LLMs. Many models demonstrate reasonable performance on MC, yet their accuracy drops significantly when handling OE or FB questions, which require

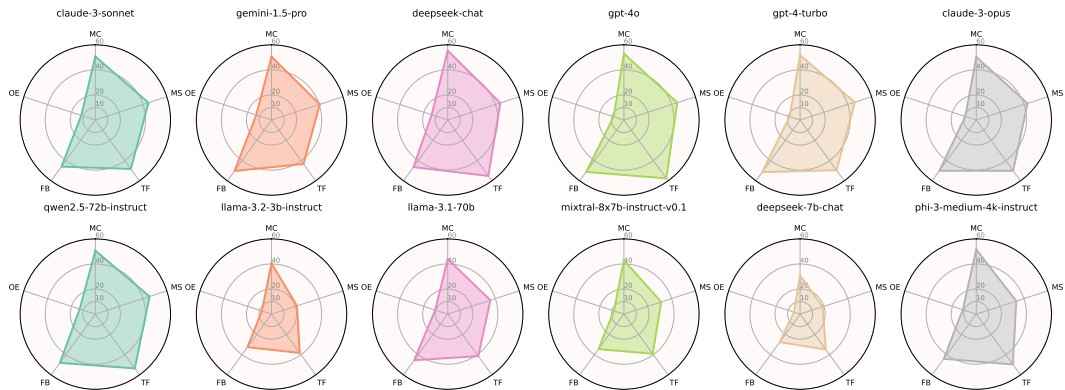

Figure 6: Radar diagrams reflecting the mastery of LLMs over various question types, the first row of six LLMs are for closed-source ones while the second row consists of the open-source ones.

.

deeper reasoning and articulation. For example, Qwen2.5-72b, which performs strongly on MC questions, struggles when tasked with non-MC questions, reflected in its lower Q→AR scores. Additionally, the poor performance on MS questions suggests another problem. MS questions demand more than recognizing a pattern among predefined choices; it requires the model to generate coherent, contextually accurate based on the anti-disturbing ability. This additional complexity likely contributes to the sharp decline in performance observed across models when addressing these types of questions. The results demonstrate that current LLMs are not yet capable of the nuanced reasoning and articulation required for more complex college-level tasks.

**Model size does not inherently guarantee the reasoning ability**. Our experimental results in Table 4 challenge the assumption that larger models inherently possess superior reasoning abilities. In this table, all the LLMs are categorized into corresponding model families. While larger models like Qwen2.5-72b-Instruct and GPT-4-turbo excel in Q→A performance, their Q→AR scores reveal that size alone does not guarantee an understanding of underlying logic, illustrating that even the largest models still exhibit another strong limitation in their ability to reason and articulate rationales.

More convincing examples can be found in the table. Among the Qwen model family, while qwen2.5-32b-instruct falls behind qwen2.5-72b-instruct in terms of Q→A with 0.74% accuracy difference, it surprisingly surpasses the 72b version with 1.5% improvement on Q→AR. Another example is in the Llama model family where llama-3-8b-instruct could achieve a comparable Q→AR performance with llama-3.1-70b-instruct. Notably, the differences between their Q→A performance is about 10.07% while the gap is merely 2.33% on Q→AR. These observations clearly suggest that while the model size indeed plays a substantial role in direct answer prediction, i.e., Q→A, it does not directly correlate with a model's reasoning ability, i.e., Q→AR.

**Discipline-specific insights**: ($i$) Context-intensive tasks. Topics such as AI Introduction and Ethics of CS and AI require a solid understanding of conceptual frameworks. All three state-of-the-art LLMs demonstrate a surprisingly low performance with around 12.5% and 6.5% Q→AR. This suggests challenges in grasping nuanced ethical considerations. Similarly, scores in the AI Introduction domain reflect the models' difficulty in conveying complex ideas coherently. These results highlight that while LLMs possess substantial factual knowledge, they often struggle with articulating and rationalizing that knowledge in contextually rich environments. ($ii$) Reasoning-intensive tasks. Mathematics and Computer Architecture reflect a different situation compared with context-intensive tasks. All three models could perform significantly satisfying the Q→A criteria, however, their performance decreases sharply on Q→AR with over 30% accuracy drop in average. This reflects the necessity for improved training techniques that focus on fostering deeper cognition for reasoning-intensive tasks.

## 5    RELATED WORK

In recent years, the development of various benchmarks has significantly improved the evaluation of large language models (LLMs) from multiple task perspectives. General benchmarks like

MMLU (Hendrycks et al., 2020), MMLU-Pro (Wang et al., 2024a), GLUE (Wang et al., 2021), SuperGLUE (Sarlin et al., 2020), CommonSenseQA (Talmor et al., 2018), BIG-Bench (Srivastava et al., 2022), and the AI2 Reasoning Challenge (ARC-Challenge) (Clark et al., 2018) have played a pivotal role in advancing the understanding of language, reasoning, and question-answering tasks, leading to more specialized and comprehensive assessments. More recent benchmarks, such as BoolQ (Clark et al., 2019), HellaSwag (Zellers et al., 2019), and SQuAD (Rajpurkar et al., 2018), further enhance the evaluation of model comprehension and complex reasoning abilities. Task-specific benchmarks like HumanEval (Chen et al., 2021), MBPP (Austin et al., 2021), GSM8K (Cobbe et al., 2021), and MATH (Hendrycks et al., 2021) have broadened the evaluation landscape, covering diverse areas and expanding the scope of LLM evaluation.

To facilitate performance comparisons between LLMs, platforms like the OpenLLM Leaderboard (Aidar Myrzakhan, 2024) and OpenCompass (Contributors, 2023) have been established. However, as LLMs evolve rapidly, leaderboard scores are becoming increasingly saturated at the top. Models such as GPT-4 (Achiam et al., 2023), Claude-3 (Anthropic, 2023b), and open-source models like the Qwen (Yang et al., 2024) series are achieving near-perfect results across various benchmarks. This trend underscores the need for more challenging benchmarks to push the limits of LLM capabilities. Moreover, existing benchmarks typically focus solely on reporting the evaluation results of LLMs, limiting assessments to whether an answer is correct while overlooking the reasoning process and rationale that reveal a model's true understanding of a question. To address this gap, we introduce a multi-type rationalized dataset featuring 1,018 questions across five types and 16 subjects in computer science and AI, designed to challenge LLMs with college-level reasoning tasks. For a more robust evaluation, we propose two new metrics: Q→A and Q→AR, which assess both the correctness of answers and the quality of their rationales.

Our findings show that LLMs tend to guess college-level answers without fully grasping the rationale behind them. Interestingly, larger models do not consistently outperform smaller ones in terms of reasoning. Through extensive experiments involving 40 LLMs, we provide valuable insights into performance variations across question types, topics, and model sizes.

## 6 CONCLUSION AND FUTURE WORK

In this paper, we present `CLR-Bench`, a novel benchmark specifically designed to evaluate the reasoning capabilities of large language models in college-level tasks, focusing on computer science and artificial intelligence. We release a high-quality multi-type question-answering dataset comprising 1,018 questions spanning 16 disciplines. A novel reasoning evaluation paradigm is proposed through Q→A and Q→AR metrics. Unlike traditional benchmarks that solely assess the correctness of final answers, our framework goes beyond by requiring models to provide coherent rationales for their answers, ensuring a deeper evaluation of their reasoning capabilities.

Through extensive experiments on 40 LLMs, we observed significant gaps between the accuracy of answers (Q→A) and the combined performance on answers and rationales (Q→AR). Our key insights include: ($i$) LLMs tend to '**guess**' the answers since higher Q→A often fails to lead to higher Q→AR. Even when models achieve high accuracy on answers alone, their Q→AR scores were notably lower, indicating that models often fail to fully understand the rationale behind their correct answers. This observation underscores the need for better reasoning mechanisms within LLMs and suggests that current models may rely on shortcuts or superficial patterns rather than truly grasping the underlying concepts. $ii$ Model size does not consistently guarantee a better reasoning ability. Smaller models may even surpass larger ones in terms of Q→AR, even though they fall behind on Q→R. We believe this observation may inspire the community for further exploration toward achieving a more robust understanding beyond the direct answer prediction.

In future work, our benchmark will continue expanding to more college-level disciplines including law, physics, accounting, chemistry, biology, etc, and include more advanced techniques and LLMs to deeply investigate the development of research on improving reasoning abilities.

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

## A  REPRODUCIBILITY AND OPEN-SOURCE DATASET

To ensure the usability and reproducibility of our benchmark and dataset, we have open-sourced both the dataset and codes at `https://anonymous.4open.science/r/CLR-Bench-7771`.

Specifically, we include five separated JSONL files for MC, MS, TF, FB, and OE questions, respectively. The automatic rationale generation, as well as the evaluator for five types of prediction and rationales, are also summarized in the folder.

## B  ADDITIONAL EXPERIMENTAL ANALYSIS

In this section, we showcase the overall performance comparison of Q→A and Q→AR in Table 4, where all the LLMs are categorized into model families.

In general, we group the models into Claude, GPT, Deepseek, Gemma, Llama, Mistral, Phi, Qwen, and Yi. Sufficient cases demonstrated that under the same model architecture, though smaller modes always fall behind the larger ones due to the model size, they could sometimes surpass in terms of the reasoning ability with comparable or even higher Q→AR scores.

Table 4: `CLR-Bench` evaluation on different LLM families.

| # | LLM Families | Q→A | Q→AR |
|---|---|---|---|
| 1 | claude-3-opus | 62.57% | 38.56% |
| 2 | claude-3-sonnet | 60.51% | 38.51% |
| 3 | gpt-3.5-turbo | 53.98% | 34.70% |
| 4 | gpt-4-turbo | 63.31% | 39.00% |
| 5 | gpt-4o | 60.76% | 41.60% |
| 6 | deepseek-chat | 62.43% | 42.09% |
| 7 | deepseek-7b-base | 40.08% | 20.73% |
| 8 | deepseek-7b-chat | 38.02% | 23.67% |
| 9 | gemma-7b-it | 25.83% | 18.74% |
| 10 | gemma2-9b | 16.26% | 8.77% |
| 11 | gemma2-9b-it | 53.54% | 34.63% |
| 12 | llama-2-7b-chat | 35.07% | 21.32% |
| 13 | llama-2-7b | 38.75% | 20.43% |
| 14 | llama-3-8b | 46.61% | 24.19% |
| 15 | llama-3-8b-instruct | 50.15% | 31.34% |
| 16 | llama-3.1-8b | 48.82% | 26.11% |
| 17 | llama-3.1-8b-instruct | 50.49% | 29.54% |
| 18 | llama-3.1-70b | 56.97% | 33.60% |
| 19 | llama-3.1-70b-instruct | 60.22% | 33.67% |
| 20 | llama-3.2-1b-instruct | 33.10% | 19.38% |
| 21 | llama-3.2-3b-instruct | 43.37% | 28.36% |
| 22 | mistral-7b-instruct-v0.1 | 43.27% | 26.28% |
| 23 | mistral-7b-v0.1 | 48.18% | 26.33% |
| 24 | mixtral-8x7b-instruct-v0.1 | 41.36% | 30.48% |
| 25 | mixtral-8x7b-v0.1 | 51.38% | 29.00% |
| 26 | phi-3-mini-4k-instruct | 53.78% | 35.56% |
| 27 | phi-3-medium-4k-instruct | 57.12% | 37.60% |
| 28 | qwen1.5-7b-chat | 26.67% | 22.35% |
| 29 | qwen1.5-7b | 47.10% | 27.16% |
| 30 | qwen2.5-7b | 54.62% | 33.37% |
| 31 | qwen2.5-7b-instruct | 56.93% | 37.25% |
| 32 | qwen2.5-32b-instruct | 63.31% | **42.29%** |
| 33 | qwen2.5-72b | 62.52% | 37.89% |
| 34 | qwen2.5-72b-instruct | **64.05%** | 40.79% |
| 35 | yi-1.5-6b-chat | 41.60% | 25.56% |
| 36 | yi-1.5-6b | 48.08% | 27.80% |
| 37 | yi-1.5-34b-chat | 52.95% | 33.87% |
| 38 | yi-1.5-34b | 56.43% | 32.22% |

## C  EXPERT-GUIDED PROMPT TEMPLATE FOR GOLD RATIONALE GENERATION

**Multi-Choice Question (MC):** The rationale should justify the one selected answer, explaining why this answer is the correct one, and how the other options fall short.

> Given this multiple-choice question: `Q` and corresponding choices: `A`, explain why the choice: `c_best` is the correct answer while the others are incorrect, including necessary explanations and relevant information. Make it less than forty words.

**Multi-Select Question (MS):** The rationale should justify all selected answers, explaining why each is correct, and how the other options fall short.

> Given this multi-select question: `Q` and corresponding choices: `A`, explain why the choices: `c_bests` are the correct answers while the others are incorrect, including necessary explanations and relevant information. Make it less than forty words.

**True-or-False Question (TF):** The rationale needs to support the truth value of the statement with concise reasoning based on factual knowledge.

> Given this True-or-False question: `Q`, explain why this statement: `s` is True (or False), including necessary explanations and relevant information. Make it less than twenty words.

**Fill-in-the-Blank Question (FB):** The rationale should explain why the inserted term or phrase is the most contextually appropriate, with a brief justification.

> Generate a rationale for this fill-in-blank question: `Q`, including necessary explanations and relevant information. Make it less than forty words.

**Open-Ended Question (OE):** For open-ended questions, the rationale must cover the expected knowledge points and explain the core components of a satisfactory answer.

> Generate a rationale for this open-ended question: `Q`, indicate what kind of answer is satisfactory, including the coverage of knowledge points, necessary reasoning steps, well-rounded explanations, and relevant information. Make it less than forty words.

## D  PROMPT TEMPLATES FOR OPEN-ENDED ANSWER AND RATIONALE EVALUATION

**Evaluation Prompt for Rationale.** The assessment of the generated rationale will be presented through a tiered scoring based on the quality of the rationale, which includes three distinct categories: 0, 0.5, and 1. Further details regarding the prompt template are outlined below:

> You are a strict evaluator. Compare the following two rationales for correctness and completeness:
> Predicted Rationale: **[pred_rationale]**
> Gold Rationale: **[gold_rationale]**
> Please evaluate the predicted rationale in comparison to the gold rationale. Respond with a score between 0 and 1:
> - 1: The predicted rationale fully aligns with the gold rationale.
> - 0.5: The predicted rationale is partially correct but lacks completeness or includes incorrect information.
> - 0: The predicted rationale is incorrect or completely misaligned with the gold rationale.
> only provide the score without any explanation.

**Evaluation Prompt for Open-Ended Answer.** Given that the open-ended response is inherently text-based, the evaluation prompt template of the open-ended answer is designed as follows:

> You are a strict evaluator. Compare the following two answers for correctness and completeness:
> Predicted Answer: **[pred_answer]**
> Gold Answer: **[gold_answer]**
> Please evaluate the predicted answer in comparison to the gold answer. Respond with a score between 0 and 1:
> - 1: The predicted answer fully aligns with the gold answer.
> - 0.5: The predicted answer is partially correct but lacks completeness or includes incorrect information.
> - 0: The predicted answer is incorrect or completely misaligned with the gold answer.
> only provide the score without any explanation.

## E    CASE STUDIES OF QUESTIONS IN CLR−BENCH

In this section, we present a series of case studies that exemplify the diverse types of questions included in the dataset. Moreover, we showcase another set of examples that illustrate GPT-4o may potentially guess the correct answer but offer incorrect or incomplete rationales, suggesting that these models often rely on heuristic guessing based on context and choices rather than deep understanding.

### E.1    EXAMPLES OF MULTI-TYPE QUESTIONS

In this subsection, we provide some question examples of each question type as follows:

(i) Multi-choice Question Example:

> **Question**: "Which of the following best describes the concept of monotonicity in a knowledge base?"
> **Choices**: "A": "New assertions can invalidate previous conclusions.", "B": "Inference rules can be applied only to current premises.", "C": "Inference rules are always valid regardless of additional information.", "D": "Adding new information can alter previously drawn conclusions."
> **Topic**: "Artificial intelligence introduction", "Level-2 Topic": "Knowledge base"
> **Answer**: "C"
> **Rationale**: "Monotonicity ensures that adding new assertions to a knowledge base does not invalidate previously inferred conclusions, as inference rules remain valid regardless of additional information."

(ii) Multi-select Question Example:

> **Question**: "Which techniques are used to analyze malware behavior?"
> **Choices**: "A": "Static analysis", "B": "Dynamic analysis", "C": "Behavioral analysis", "D": "Network analysis"
> **Topic**: "Cybersecurity", "Level-2 Topic": "Malware analysis"
> **Rationale**: "Static, dynamic, and behavioral analyses are essential techniques for understanding malware functionality, identifying patterns, and assessing its impact on systems."
> **Answer**: "ABC"

(iii) True-of-false Question Example:

> **Question**: "A memory hierarchy provides unlimited fast memory as desired by programmers."
> **Topic**: "Computer architecture", "Level-2 Topic": "Memory hierarchy"
> **Answer**: "False"
> **Rationale**: "The memory hierarchy addresses the limitations of memory technologies by leveraging varying speeds and costs, enabling efficient data access while managing the practical constraints of memory availability and performance."

(iv) Fill-in-blank Question Example:

> **Question**: "The _______ component of tf-idf measures how frequently a term appears in a specific document."
> **Topic**: "Information retrieval", "Level-2 Topic": "Tf-idf"
> **Answer**: "term frequency"
> **Rationale**: "Term frequency quantifies the occurrence of a term within a document, influencing its weight in the tf-idf calculation and overall relevance in search results."

(v) Open-Ended Question Example:

> **Question**: "Discuss why each input port in a high-speed router stores a shadow copy of the forwarding table."
> **Topic**: "Computer networks", "Level-2 Topic": "Routing and switching"
> **Answer**: "With the shadow copy, the forwarding lookup is made locally, at each input port, without invoking the centralized routing processor. Such a decentralized approach avoids creating a lookup processing bottleneck at a single point within the router."
> **Rationale**: "Each input port in a high-speed router stores a shadow copy of the forwarding table to enable local, decentralized lookups. This approach prevents bottlenecks at the centralized routing processor, ensuring faster and more efficient packet forwarding."

### E.2    EXAMPLES OF GUESSING BEHAVIORS

**(i) Multi-Choice Question Example:**

**Question**: "An entity (E) has 5 attributes and 3 of them form the composite primary key. The entity has a recursive 1-to-many relationship. How many attributes should there be after transforming the entity (E) to the corresponding table?" "predict_rationale": "The table will have 8 attributes after transforming the entity, but none of the provided options match this number."

**Choices**: "A": "4", "B": "5", "C": "6", "D": "None of the above"

**Topic**: "Database systems"

**Answer**: "D"

> **Predicted Rationale**: "The table will have 8 attributes after transforming the entity, but none of the provided options match this number."
> **Gold Rationale**: "Transforming to a table retains all 5 attributes. Recursive relationships don't add attributes to the table schema."

**Question**: "An entity (E) has 5 attributes and 3 of them form the composite primary key. The entity has a recursive 1-to-many relationship. How many attributes should there be after transforming the entity (E) to the corresponding table?"

**Choices**: "A": "4", "B": "5", "C": "6", "D": "None of the above""

**Topic**: "Cybersecurity", "Level-2 Topic": "Entity to table transformation"

**Answer**: "D"

> **Predicted Rationale**: "SSL/TLS is a cryptographic protocol designed to secure data in transit, while FTP, HTTP, and Telnet do not provide encryption and are therefore insecure for transmitting sensitive information."
> **Gold Rationale**: "Firewalls block unauthorized access by filtering incoming and outgoing traffic based on predetermined security rules, helping to protect networks from intrusions and attacks."

**(ii) True-or-flase Question Example:**

**Question**: "In the step of preprocessing, tuple duplication is not needed at all." (Incomplete rationale)

**Topic**: "Data science and big data", "Level-2 Topic": "Data mining"

**Answer**: "False"

> **Predicted Rationale**: "Preprocessing often involves cleaning and preparing data for analysis, which can include removing duplicates to ensure data quality and consistency. However, there are scenarios where tuple duplication might be necessary, such as when creating synthetic data for testing or when specific analyses require repeated data points. Therefore, the necessity of tuple duplication depends on the context and goals of the preprocessing step."
> **Gold Rationale**: "Tuple duplication is a part of data integration."

**Question**: "Suppose an attacker collects a large amount of initial sequence numbers sent by the server. Can the attacker cause the server to create many fully open connections by sending ACKs with those initial sequence numbers?" (Incomplete rationale)

**Topic**: "Computer networks", "Level-2 Topic": "Network security"

**Answer**: "False"

> **Predicted Rationale**: "Initial Sequence Numbers (ISNs) are used in the TCP three-way handshake to establish a connection. The handshake involves the following steps: the client sends a SYN packet with an ISN, the server responds with a SYN-ACK packet containing its own ISN, and the client completes the handshake with an ACK packet. If an attacker collects ISNs sent by the server, they could potentially send ACK packets with those ISNs to trick the server into thinking a connection has been established. However, for a connection to be fully open, the server expects a valid SYN packet from the client first. Simply sending ACKs with collected ISNs without the initial SYN packet would not complete the three-way handshake, and thus would not create fully open connections."
>
> **Gold Rationale**: "The sequence numbers also have a timestamp and a time for which it will be active."

**(iii) Fill-in-blank Question Example:**

**Question**: "If an array has 100 elements, then KnuthŽ2019s algorithm would start with an interval of ________"

**Topic**: "Data structures and algorithms", "Level-2 Topic": ""

**Answer**: "40"

> **Predicted Rationale**: "Knuth's algorithm is a specific sequence used in Shell sort, a generalization of insertion sort. The sequence is defined as $h = 3h + 1$, starting from $h = 1$. For an array of 100 elements, we need to find the largest interval $h$ that is less than 100. Starting from $h = 1$, we get the sequence: 1, 4, 13, 40, 121, etc. The largest interval less than 100 is 40."
>
> **Gold Rationale**: "Knuth's algorithm initializes the interval as half the array size, so for 100 elements, it starts with an interval of 50. However, since the interval is halved each iteration, starting with 40 ensures it covers the entire array."

**(iv) Open-ended Question Example:**

**Question**: "In a complete binary tree with 20 nodes, and the root considered to be at level 0, how many nodes are there at level 4?" (Incomplete rationale)

**Topic**: "Data structures and algorithms", "Level-2 Topic": "Data structure"

**Answer**: "5"

> **Predicted Rationale**: "In a complete binary tree, each level doubles the number of nodes. With 20 nodes, level 4 has 5 nodes because $2^4 = 16$ and the last 4 nodes are distributed across levels 0-3."
>
> **Gold Rationale**: "By calculating the total number of nodes in the levels before level 4 and subtracting from the total number of nodes in the tree, we determine the number of nodes at level 4."

