# OpenReview forum: "$\texttt{CLR-Bench}$: Evaluating Large Language Models in College-Level Reasoning"
_ICLR.cc/2025/Conference — Submitted to ICLR 2025_

### Official Review · Reviewer_MXSi · 2024-10-21

**Soundness:** 3
**Presentation:** 3
**Contribution:** 2
**Rating:** 5
**Confidence:** 5

**Summary:**

This work presents CLR-BE, a dataset consisting of questions from 16 challenging college disciplines in computer science and artificial intelligence. The dataset contains 5 types of questions, while each question is associated with detailed explanations from experts. The paper introduces a method to judge the models’ reasoning abilities based on the correctness of their reasoning paths.

**Strengths:**

1.The dataset provides a new way to evaluate the reasoning abilities of LLMs, focusing on the correctness of the reasoning process instead of only on the final answer

2.The paper is well presented, with clear figures and illustrations

**Weaknesses:**

1.The domain of the problems seems limited, as it seems to only contain computer science related questions. Computer science seems not to be a very representative field for the evaluation of reasoning abilities. The dataset can benefit from inclusion of more challenging and reasoning intensive tasks from subjects such as math and physics.

2.The evaluation of the correctness of reasoning processes involves semantic similarity and GPT-4-assisted expert evaluation. I am not so sure about the accuracy of this evaluation process as reasoning paths can be varied. Also, the errors in reasoning paths can be subtle, and this method of evaluation seems a little coarse to me.

**Questions:**

The dataset contains a lot of choice questions. Perhaps a more efficient way is to transform choice questions into open-ended ones with more possible final answers to ensure that a correct final answer indicates a correct reasoning process?

It would be good to have more case studies about various model generated solutions, including false positives, false negatives, ect.

---

> ### Author Response · Authors · 2024-11-23
> **Author Responses to Reviewer MXSi**
>
> Dear Reviewer MXSi,
>
> We would like to sincerely thank you for your recognition of our contributions and novelty in evaluating LLMs. We value your insightful comments (``hurray! for new findings``) and will carefully revise the final version following your suggestions.
> We also wish to invite you to check our **new** case studies following your suggestions on the experiments.
> > Responses to Weaknesses
> * **W1: Domain Limitations**. Thanks for raising the discussion and providing constructive suggestions. On one hand, we will change our title to indicate the clear domain with CS and AI to address your concerns. On the other hand, we highly value the CS and AI domain more than all the existing benchmarks for four reasons.
> **First**, CS and AI naturally contain a variety of types of challenging tasks compared with all the existing 'single-type' benchmarks. In CS and AI domain, we have **``conceptual understanding``** tasks (algorithms, networks, databases, paradigms, machine learning, NLP, CV), **``Math``** problems (Discrete mathematics, linear algebra, matrix computation, calculus), **``Code generation``** problems (programming, optimization, code completion, error detection), etc. **Second**, the performance of LLMs on CS and AI is still very unsatisfactory or even bad It is valuable to benchmark and research. **Third**, it remains a blank in the benchmark of LLMs considering CS and AI. This also brings two opportunities. **(i)** LLMs will have few chances to be fine-tuned based on our questions. **(ii)** Following research on RAG or LLMs themselves could benefit a lot from our benchmark. **Forth**, we are pioneering in a more comprehensive one in terms of **multiple questions types** and the **difficulty in college-level**. The real scenarios of applying LLMs could be much more complicated while single-type questions (in existing benchmarks) could hardly satisfy.
> * **W2: Quality of evaluation**. We understand your concerns which is also our concern in the very beginning. Following the widely adopted way of ``LLM-as-a-judge``[1], e.g., OlympicArena[2] and Infimm-Bench[3], we employ GPT-4 as the base referees scoring the rationale among {0,0.5,1}. **What's more**, in this paper, we propose a paradigm for more reliable expert-enhanced auto-evaluation for different types of questions. We carefully introduce the expertise to constrain the marking criteria for different types of questions through elaborate prompts and instruct GPT-4 to score the rationale. This could effectively **``align``** the auto-evaluation with human experts.
>
> > Responses to questions
> * **Q1: Transforming MC to OE**. We would like to say hurray for your valuable suggestion! Following this, we have conducted a new round of case selection. An interesting observation is that there is no OE questions are answered wrongly with correct rationale. We can temporarily say that in college-level complex reasoning tasks, OE questions will to some extent ensure the reasoning process by only evaluating the answer's correctness. However, we do believe that the MC, MS, TF and FB questions have their indispensable merits that could support our findings that LLMs tend to 'guess' the answers according to the context. We will collect more OE questions and conduct more experiments following your suggestions.
> * **Q2: More case studies**. Thank you for your constructive comment. We have included more case studies with partially correct rationale but wrong answers for your reference hereunder.

---

> > ### Author Response · Authors · 2024-11-23
> > **More case studies following Reviewer MXSi's suggestion (a)**
> >
> > FB:
> > ```python
> > {"Question":
> > "Shifting a group of items left or right requires repeated __________.",
> > "Level-1 Topic": "Programming fundamentals",
> > "Level-2 Topic": "Operator"}
> > {
> > "prediction": """RATIONALE: Shifting a group of items left or right
> > typically involves moving each item to the position of its neighbor.
> > This process is often referred to as \"rotation\" in computer science
> > and data structures, where elements are cyclically moved within an
> > array or list. Repeated rotations ensure that each element is shifted
> > to its new position. ANSWER: rotations""",
> > "gold answer": "copies"
> > }
> > ```
> > MC:
> > ```python
> > {"Question":
> > "What is the height (depth) of a complete binary tree with 2011 nodes?",
> > "Choices": {"A": "11", "B": "10", "C": "20", "D": "2011"},
> > "Level-1 Topic": "Data Structure and Algorithm",
> > "Level-2 Topic": "Data structure"}
> > {
> > "prediction": """To determine the height (or depth) of a complete binary
> > tree with 2011 nodes, we need to understand the properties of a complete
> > binary tree. In a complete binary tree, all levels are fully filled except
> > possibly the last level, which is filled from left to right.\n\nThe height
> > \\( h \\) of a complete binary tree with \\( n \\) nodes can be approximated
> > using the formula:\n\\[ h \\approx \\log_2(n) \\]\n\nGiven \\( n = 2011 \\):
> > \n\\[ h \\approx \\log_2(2011) \\]\n\nUsing a calculator or logarithm table:
> > \n\\[ \\log_2(2011) \\approx 10.97 \\]\n\nSince the height of a tree must be
> > an integer, we round up to the next whole number:\n\\[ h = 11 \\]\n\nNow,
> > let's evaluate the options:\n- A) 11: This is the correct height as calculated.
> > \n- B) 10: This is incorrect because \\( \\log_2(2011) \\) is slightly less
> > than 11, but we need to round up to the next whole number.\n- C) 20: This is
> > incorrect because it is much larger than the calculated height.\n- D) 2011:
> > This is incorrect because it represents the number of nodes, not the height.
> > \n\nRATIONALE: The height of a complete binary tree with 2011 nodes is
> > approximately 11, as calculated using the logarithmic formula. ANSWER: A""",
> > "gold": "B"
> > }
> > ```
> > MS:
> > ```python
> > {"Question":
> > """Which of the following statements about protection and instruction set
> > architecture are true? (Select all that apply)""",
> > "Choices": {"A": """Virtual memory requires changes to existing instruction
> > set architectures.""",
> > "B": "The IBM 370 instruction set was unchanged from the IBM 360.",
> > "C": "Adjustments are being made today for virtual machines.",
> > "D": "Protection is solely the responsibility of the operating system."},
> > "Level-1 Topic": "Computer architecture",
> > "Level-2 Topic": "Instruction set architecture"}
> > {
> > "prediction": """RATIONALE: Virtual memory does not necessarily require
> > changes to existing instruction set architectures, as it can be managed by
> > the operating system and hardware support. The IBM 370 instruction set was an
> > extension of the IBM 360, not unchanged. Adjustments for virtual machines
> > are indeed being made today to support efficient virtualization. Protection
> > involves both the operating system and hardware mechanisms, not solely the
> > operating system. ANSWER: BC""",
> > "gold": "AC"
> > }
> > ```

---

> > > ### Author Response · Authors · 2024-11-23
> > > **More case studies following Reviewer MXSi's suggestion (b)**
> > >
> > > OE:
> > > ```python
> > > {"Question":
> > > """As other items are inserted and removed, does a particular item in a
> > > queue\nmove along the array from lower to higher indices, or higher to lower?",
> > > "Level-1 Topic": "Data structures and algorithms""",
> > > "Level-2 Topic": "Data structure"}
> > > {
> > > "prediction": """To determine how a particular item in a queue moves along
> > > an array, we need to understand the basic operations of a queue and how it
> > > is typically implemented using an array.\n\nA queue is a First-In-First-Out
> > > (FIFO) data structure, meaning that the first element added to the queue will
> > > be the first one to be removed. The primary operations for a queue are:\n1.
> > > **Enqueue**: Adding an item to the end of the queue.\n2. **Dequeue**:
> > > Removing an item from the front of the queue.\n\nWhen a queue is
> > > implemented using an array, two indices are typically maintained:\n-
> > > **Front**: Points to the first element in the queue.\n- **Rear**: Points
> > > to the last element in the queue.\n\nHere’s how the operations affect these
> > > indices:\n- **Enqueue**: The `rear` index is incremented to add a new item
> > > to the end of the queue.\n- **Dequeue**: The `front` index is incremented
> > > to remove the item from the front of the queue.\n\nGiven this, let's consider
> > > the movement of a particular item in the queue:\n- When an item is enqueued,
> > > it is placed at the position indicated by the `rear` index.\n- As items are
> > > dequeued, the `front` index moves forward, effectively moving the position
> > > of the item towards higher indices in the array.\n\nTherefore, as other items
> > > are inserted (enqueued) and removed (dequeued), a particular item in the queue
> > > will move from lower to higher indices in the array.\n\nRATIONALE: In a queue
> > > implemented using an array, the `front` index is incremented during dequeue
> > > operations, causing items to move towards higher indices as they remain in
> > > the queue. ANSWER: A particular item in a queue moves along the array from
> > > lower to higher indices.""",
> > > "gold": "It doesn't move at all."
> > > }
> > > ```
> > > TF:
> > > ```python
> > > {"Question": """When you delete an item from an unordered array, in
> > > most cases you shift other items to fill in the gap.""",
> > > "Level-1 Topic": "Data structures and algorithms",
> > > "Level-2 Topic": "Algorithm design techniques"}
> > > {
> > > "prediction": """RATIONALE: In an unordered array,
> > > when an item is deleted, the typical approach to maintain the array's
> > > structure is to replace the deleted item with the last item in the array
> > > and then reduce the array's size by one. This method avoids the need to
> > > shift multiple elements, which would be necessary in an ordered array to
> > > maintain order. Therefore, in most cases, you do not shift other items to
> > > fill in the gap when deleting from an unordered array. ANSWER: False""",
> > > "gold": "True"
> > > },
> > > ```
> > >
> > > [1]Judging llm-as-a-judge with mt-bench and chatbot arena. Lianmin Zheng, et al. NeurIPS 2024
> > > [2]OlympicArena: Benchmarking Multi-discipline Cognitive Reasoning for Superintelligent AI. Zhen Huang, et al. NeurIPS 2024
> > > [3]InfiMM-Eval: Complex Open-Ended Reasoning Evaluation For Multi-Modal Large Language Models. Xiaotian Han, et al. arxiv 2023.

---

> > > > ### Author Response · Authors · 2024-12-02
> > > > **indly Requesting Your Feedback on Our Rebuttal**
> > > >
> > > > Dear Reviewer witE,
> > > >
> > > > We would like to use this final opportunity to kindly ask for your reconsideration of the advantages of CS/AI and the quality of evaluation. We also value your insightful comments on the experiments and case studies, they have indeed improved the quality of our paper a lot.
> > > >
> > > > As the deadline for the Rebuttal approaches, we want to inquire whether our responses have addressed your concerns. If any remaining problems or areas require further clarification, we would greatly appreciate it if you could provide us with additional feedback.
> > > >
> > > > Best,
> > > >
> > > > Authors of Paper 5453

---

### Official Review · Reviewer_witE · 2024-10-26

**Soundness:** 2
**Presentation:** 2
**Contribution:** 2
**Rating:** 3
**Confidence:** 5

**Summary:**

This paper proposes a novel benchmark CLR-Bench to evaluate the reasoning ability of LLM in college-level **computer science and artificial intelligence** tasks.
Based on the experimental results, the authors found that current LLMs tend to ‘guess’ answers for college-level questions, i.e., LLMs can reach the correct answer but contains incorrect reasoning steps.

**Strengths:**

- a novel benchmark on computer science and artificial intelligence tasks is proposed
- the writing is easy to follow

**Weaknesses:**

- many wrong statements in this paper try to mislead readers:
  - In Table 1, GSM8K and MATH are not multiple-choice questions. Moreover, their samples contain complex rationale to evaluate LLMs.
  - In L105, the authors claim that $Q\to A$ and $Q \to AR$ are two novel metric proposed by them. however, $Q\to A$ (i.e., outcome accuracy) has been widely used in many reasoning tasks to evaluate the performance. also, $Q\to R$ is equivalent to the process accuracy proposed in the paper Let's verify step by step (https://arxiv.org/abs/2305.20050). The only minor novelty in this paper is the $Q\to AR$ metric, which assigns 0.5 to the prediction if the answer is wrong but its process is correct.
- evaluating the correctness of the reasoning path is a popular topic in LLM (some related works are list below), however, none of them are discussed in this paper.
  - Evaluating Mathematical Reasoning Beyond Accuracy
  - SELF-[IN]CORRECT: LLMs Struggle with Discriminating Self-Generated Responses
  - SELFCHECK: USING LLMS TO ZERO-SHOT CHECK THEIR OWN STEP-BY-STEP REASONING
  - Let's Verify Step by Step
  - The Generative AI Paradox: “What It Can Create, It May Not Understand”
- the proposed benchmark is very small (only about 1k samples) and is very narrow (computer science). Hence, the observations are also limited to the computer science domain and cannot be generalized to other reasoning tasks.
- Previous works usually study the process correctness of mathematical problems. compared with existing math benchmarks, what are the essential advantages of using computer science discipline?

**Questions:**

see above Weaknesses

---

> ### Author Response · Authors · 2024-11-23
> **Author Responses to Reviewer witE**
>
> Dear Reviewer witE,
>
> We would like to gratefully thank you for your carefulness and constructive comments. All your suggestions and references will be included and well-cited in the revision.
> We also sincerely wish to invite you to check our detailed rebuttal to address your concerns and prove our merits to the community, as well as our nonnegligible contributions in benchmarking LLMs, including the novel and systematic dataset construction based on hierarchical topic graphs, careful expert-enhanced benchmark evaluation and the unique value of CS and AI in college-level.
>
> > Responses to Weaknesses
> * **W1: Wrong statements**. Thank you for your carefulness about our typos. We will correct them in the final version. In Table1, GSB8K and MATH were originally supposed to be 'OE'. This results in no misleading since we wanted to highlight that all the existing benchmarks are **``Single-type``** datasets while our datasets contain **``FIVE``** types of equally distributed questions to comprehensively evaluate the LLMs across different answering scenarios.
> For metrics, we will revise the expression and only keep Q->AR as the newly proposed ones. Again, no misleading since our intention was to excitedly share the observations with the community by comparing Q->A and Q->AR, LLMs are showing a dramatic decrease indicating a 'guessing' behavior. We will also properly cite 'Let's verify step by step'. Finally, we would like to clarify that Q->AR is **not** 'assigning 0.5 to the prediction if the answer is wrong but its process is correct'. We have introduced in Line 316 to Line 338 where our Q->AR will tend to punish the 'guessing' behavior with lower scores.
>
> * **W2: Missing references**. Thanks for pointing out these papers, we will properly cite them in the revised version. Different from research papers, we neither focus on designing various models to interfere, nor decompose the reasoning steps and fine-tune LLMs for better reasoning abilities. We would like to claim our contribution as a benchmark, we propose a novel way to construct a systematic dataset containing five types of questions (MS,MC,TF,FB,OE) and three types of tasks (conceptual understanding, Math, code generation) and a more reliable evaluation covering most LLMs.
>
>
> * **W2: Small Dataset Size**. Thanks for your comments. Indeed, as a concentrating dataset that focuses on CS and AI domain containing 16 disciplines, 1018 is already a very **``big``** number. For example, MMLU[1] is currently one of the largest LLM benchmark datasets containing 12000 questions. However, it covers 57 domains which means there are only around 210 questions are allocated for each domain. Another instance is Infi-Bench[2], which focuses on code generation and only contains 286 questions from stack overflow. Therefore, 1018 questions in our benchmark should already satisfy the requirement to comprehensively evaluate LLMs for particular abilities. We will indicate the domain in our title and storyline, and also explain why we highly value CS and AI in the next paragraph.
>
> * **W3: Domain Limitations**. Thanks for raising this insightful discussion. For sure, we will include more domains in the future work. However, in terms of 'narrow', we, in reverse, highly value this domain more than all the existing benchmarks for four reasons.
> **First**, CS and AI naturally contain a variety of types of challenging tasks compared with all the existing 'single-type' benchmarks. In CS and AI domain, we have **``conceptual understanding``** tasks (algorithms, networks, databases, paradigms, machine learning, NLP, CV), **``Math``** problems (Discrete mathematics, linear algebra, matrix computation, calculus), **``Code generation``** problems (programming, optimization, code completion, error detection), etc. **Second**, the performance on LLMs on CS and AI is still very unsatisfactory or even bad It is valuable to benchmark and research. **Third**, it remains a blank in the benchmark of LLMs considering CS and AI. This also brings two opportunities. **(i)** LLMs will have few chances to be fine-tuned based on our questions. **(ii)** Following research on RAG or LLMs themselves could benefit a lot from our benchmark. **Forth**, we are pioneering in a more comprehensive one in terms of **multiple questions types** and the **difficulty in college-level**. The real scenarios of applying LLMs could be much more complicated while single-type questions (in existing benchmarks) could hardly satisfy.
>
> In general, we would like to have your inspection again please for our contributions including a novel dataset construction, benchmark evaluation and the unique value of CS and AI domain which we believe will definitely inspire the community for future benchmark and research.
>
> [1] Measuringmassivemultitask language understanding.  DanHendrycks, et al. ICLR 2021.
> [2] InfiBench: Evaluating the Question-Answering Capabilities of Code Large Language Models. Linyi Li, et al. NeurIPS 2024.

---

> > ### Comment · Reviewer_witE · 2024-11-27
> > **official comments by Reviewer witE**
> >
> > Thanks for the rebuttal.
> >
> > I am happy that the authors would fix the wrong statements/contributions, discuss important related works, and re-clarify their contributions.
> > However, I still lean toward rejection, as proposing a small dataset/benchmark on CS/AI is not enough for a top-tier conference paper.

---

> > > ### Author Response · Authors · 2024-11-28
> > > **Grateful Appreciation and further rebuttal to Reviewer witE**
> > >
> > > Dear Reviewer witE,
> > >
> > > We would like to sincerely express our gratitude for your acknowledgment and happiness with our rebuttal. However, we still value this opportunity to kindly ask for your reconsideration of our contributions and observations.
> > >
> > > > **Size of the dataset**.
> > >
> > > We totally understand your concerns about the size of a dataset. However, as a concentrated dataset focusing on one valuable domain, the size of our $\texttt{CLR-Bench}$, i.e., **1,018**,  has already satisfied the requirements to comprehensively evaluate the models' performance and make ``convincing`` and ``interesting`` observations.
> > >
> > > For your convenience and kind check, we have listed some of other concentrated benchmarks that are published on top-tier conferences, e.g., NeurIPS, with relatively small dataset sizes.
> > > 1. ``NeurIPS 2023``.  MT-bench[1], containing **80** questions only, is designed to test multi-turn conversation and instruction-following ability.
> > > 2. ``NeurIPS 2023``.  FELM [2], only contains **847** questions to evaluate the factuality of responses generated by LLMs.
> > > 3. ``NeurIPS 2023``.  PlanBench [3], only contains **885** instances to evaluate the planning and reasoning capabilities of LLMs.
> > > 4. ``NeurIPS 2024``.  Infi-Bench [4] focuses on LLM-based code generation and only contains **286** questions from stack overflow.
> > > 5. ``NeurIPS 2024``. DiReCT [5] contains **511** clinical notes to evaluate LLMs for clinical scenarios, each meticulously annotated by physicians.
> > > 6. ``NeurIPS 2024``. MedSafetyBench [6] comprises **1,800** harmful medical requests and corresponding safe responses.
> > >
> > > > **Unique and inspiring contributions**.
> > > * **Dataset construction**: we are the **``first``** to propose a comprehensive and systematic way to construct dataset based on a novel hierarchical topic graph.
> > > * **Diversity**: a) we are the **``first``** dataset containing ``5`` types of questions to evaluate LLM's reasoning ability. b) Our dataset contains a variety of types of challenging tasks including ``conceptual understanding tasks``, ``Math problems`` and ``Code generation problems``.
> > > * **Coverage**: we cover **``40``** mainstream LLMs including open-/close-source and ranging from 1b to 72b to ensure the reliability of our observations.
> > > * **Observations**: we are the **first** benchmark than obtain two interesting observations that a) LLMs tend to ‘guess’ answers according to the context. b) Model sizes do not inherently guarantee superior performance in Q→AR.
> > >
> > > [1] Judging LLM-as-a-Judge with MT-Bench and Chatbot Arena, lianmin Zheng, et al. NeurIPS 2023.
> > > [2] FELM: Benchmarking Factuality Evaluation of Large Language Models.  Shiqi Chen, et al. NeurIPS 2023.
> > > [3]  PlanBench: An Extensible Benchmark for Evaluating Large Language Models on Planning and Reasoning about Change. Karthik Valmeekam, et al. NeurIPS 2023.
> > > [4] InfiBench: Evaluating the Question-Answering Capabilities of Code Large Language Models. Linyi Li, et al. NeurIPS 2024.
> > > [5] DiReCT: Diagnostic Reasoning for Clinical Notes via Large Language Models, Bowen Wang, et al. NeurIPS 2024.
> > > [6] MedSafetyBench: Evaluating and Improving the Medical Safety of Large Language Models, Tessa Han, et al. NeurIPS 2024.

---

> > > > ### Author Response · Authors · 2024-12-02
> > > >
> > > > Dear Reviewer witE,
> > > >
> > > > Thanks again for your constructive comments and prompt feedback about our rebuttal. We value this final chance to kindly ask for your reconsideration about the size of our dataset given plenty of high-quality but small datasets and benchmarks on top-tier conferences.
> > > >
> > > > As the deadline for the Rebuttal approaches, we want to inquire whether our responses have addressed your concerns. If any remaining problems or areas require further clarification, we would greatly appreciate it if you could provide us with additional feedback.
> > > >
> > > > Best,
> > > >
> > > > Authors of Paper 5453

---

### Official Review · Reviewer_nzLV · 2024-10-31

**Soundness:** 2
**Presentation:** 2
**Contribution:** 2
**Rating:** 5
**Confidence:** 4

**Summary:**

This paper evaluates the reasoning capabilities of current LLMs, introducing a new evaluation method and corresponding dataset. The author argues that assessing only the correctness of answers is insufficient; it is also necessary to evaluate how the model arrives at these answers. To address this, the author constructed a dataset containing different types of questions and answers, along with the reasoning processes behind the answers. The study evaluated both the accuracy of the answers provided by the LLMs and the correctness of their reasoning processes. Testing on mainstream LLMs revealed that these models tend to guess the answers, as the accuracy of their reasoning processes is generally much lower than that of their answers.

**Strengths:**

1. Currently, considering only answer correctness is indeed insufficient to comprehensively evaluate the reasoning ability of LLMs. The author’s proposal to assess the correctness of the reasoning process is a valuable direction.

2. The author evaluated mainstream LLMs, covering a wide range of models.

3. Using this newly proposed evaluation method, the author arrived at different conclusions, finding that LLMs tend to guess answers, as their reasoning correctness is significantly lower.

**Weaknesses:**

1. This paper discusses a lot about the logic and process of building datasets, but it does not include any steps for ensuring or validating data quality within this process. Although the paper emphasizes the high quality of the newly proposed evaluation dataset, there is no guarantee of this quality in the outlined process. Simply assuming that human-provided annotations are of high quality is unconvincing.

2. Some descriptions in this paper lack precision. For example, in line 321, how exactly is "partially correct" determined? Does it mean that as long as the first sentence is correct, it counts as partially correct, or is it based on the subjective judgment of experts?

3. The author provides an assessment method to validate Q->R accuracy. However, this method still requires a substantial amount of manual involvement. So, how can the evaluation data proposed in this paper be better spread?

4. The author only provided the 1-shot results. Although some explanation was provided, it was not convincing to me. In my view, conducting 0-shot and k-shot validations remains an important metric for evaluating the performance of different models.

**Questions:**

Although this work provided a good point to evaluate LLMs, there are still many drawbacks as shown in weakness. Please address them accordingly.

---

> ### Author Response · Authors · 2024-11-23
> **Author Responses to Reviewer nzLV**
>
> Dear Reviewer nzLV,
>
> We would like to sincerely thank you for your recognition of the contributions and values of our work, as well as your insightful comments to improve the quality of our paper.
> We also wish to invite you to check our **new** results (zero-shot) following your suggestions on the experiments.
> > Responses to Weaknesses
> * **W1: Quality of the the dataset**. Thank you for raising the discussion. We understand your concern and are committed to addressing it showing how we **better** ensure the quality. ``Authoritative textbooks``: We adopt over 20 authoritative textbooks in 16 disciplines of CS and AI. The questions are collected from the exercises and practices, where each question is associated with a standardized reference answer and solution. This ensures our questions are well-formed and convincing for evaluating LLMs in complex college-level reasoning tasks. ``Expert-enhanced Rationale Generation``: We recruit 9 Ph.D. students to complete the rationale generation who are more convincing and well-trained than Amazon MTurk workers. **First**, given that many textbooks already provide concise rationales, experts must check and supplement necessary reasoning steps and concepts. **Second**, we ask the experts to design suitable prompts for each domain and invoke GPT-4 for the draft. **Finally**, the experts will further modify and supplement the draft based on GPT-4. This will significantly save the labor costs of experts and improve effectiveness.  ``Majority Vote-based Expert Review``. In the post-generation, we allocate three Ph.D. students to cross-check the questions and corresponding rationales. The details of the majority vote will be released in the dataset, e.g., 3/3 indicates that all experts agree with the rationale, 3/2 means two of the experts support the rationale while 1/3 means only one expert votes yes.
>
> * **W2: Definition of 'partially correct'**. Thank you for your constructive suggestions. We will make it clearer in the revised version. Following the most widely adopted ``LLM-as-a-judge``[1], e.g., OlympicArena[2] and Infimm-Bench[3], we employ GPT-4 as the base referees scoring the rationale among {0,0.5,1}. **What's more**, we carefully introduce the expertise to constrain the marking criteria for different types of questions through elaborate prompts and instruct GPT-4 to score the rationale. This could effectively **``align``** the auto-evaluation with human experts.
> * **W3: Generalization ability**. In this paper, we propose a paradigm for more reliable expert-enhanced auto-evaluation for different types of questions, e.g., MC, MS, FB, TF and OE, which is still under the mainstream of GPT-4-based ``LLM-as-a-judge``[1]. In terms of evaluation of Q->R, we could effectively **``align``** the LLM-based evaluation with human experts based on various customized prompts.
> * **W4: Zero-shot performance**. Actually, we did adopt the zero-shot setting in the very beginning. However, small-size LLMs often perform out-of-control with messy outputs. Therefore, as stated in our paper, we do a one-shot setting only for ensuring the fairness of evaluating small LLMs based on uniformity of the output. To address your concerns, we have shown the zero-shot performance of ten representative models hereunder:
>
> | #  | Models                | MC(Q->A)     | MS    | TF   | FB  | OE  | MC(Q->AR)     | MS  | TF   | FB   | OE    |
> |----|-----------------------|--------|--------|--------|--------|--------|--------|--------|--------|--------|--------|
> | 1  | gpt-4-turbo           | 78.24 | 74.56 | 61.43 | 48.38 | 42.81 | 48.34 | 42.75 | 45.56 | 49.14 | 7.12  |
> | 2  | claude-3-opus         | 76.51 | 78.03 | 59.87 | 50.67 | 41.12 | 48.92 | 41.28 | 47.26 | 48.14 | 8.43  |
> | 3  | gemini-1.5-pro        | 75.12 | 68.43 | 51.72 | 43.81 | 49.12 | 47.51 | 38.22 | 41.12 | 47.06 | 10.24 |
> | 4  | qwen2.5-7b-instruct   | 65.72 | 57.15 | 54.03 | 34.05 | 44.10 | 46.33 | 30.25 | 46.12 | 38.12 | 6.89  |
> | 5  | qwen2.5-72b-instruct  | 77.15 | 73.45 | 59.28 | 42.03 | 48.72 | 52.72 | 42.13 | 51.12 | 47.18 | 9.02  |
> | 6  | llama-3.1-8b-instruct | 57.72 | 45.05 | 48.32 | 27.82 | 39.56 | 39.28 | 24.56 | 37.06 | 33.12 | 6.78  |
> | 7  | llama-3.1-70b-instruct| 72.12 | 64.58 | 55.43 | 42.12 | 44.12 | 41.85 | 33.61 | 41.12 | 40.05 | 7.85  |
> | 8  | gemma2-9b-it          | 68.34 | 46.78 | 52.11 | 39.05 | 35.45 | 43.12 | 29.45 | 42.18 | 42.18 | 6.02  |
> | 9  | deepseek-7b-chat      | 42.19 | 21.34 | 39.12 | 17.82 | 25.71 | 28.01 | 14.23 | 29.08 | 24.15 | 3.78  |
> | 10 | yi-1.5-34b-chat       | 60.12 | 48.12 | 50.18 | 36.14 | 35.01 | 37.61 | 29.12 | 40.12 | 38.08 | 5.11  |
>
> [1]Judging llm-as-a-judge with mt-bench and chatbot arena. Lianmin Zheng, et al. NeurIPS 2024
> [2]OlympicArena: Benchmarking Multi-discipline Cognitive Reasoning for Superintelligent AI. Zhen Huang, et al. NeurIPS 2024
> [3]InfiMM-Eval: Complex Open-Ended Reasoning Evaluation For Multi-Modal Large Language Models. Xiaotian Han, et al. arxiv 2023.

---

> > ### Author Response · Authors · 2024-12-02
> > **Kindly Requesting Your Feedback on Our Rebuttal**
> >
> > Dear Reviewer nzLV,
> >
> > We would like to sincerely express our gratitude for the time and effort you have invested in reviewing our paper. Your insights and suggestions on the experiments have been invaluable in helping us improve the quality of our work. We value this final chance to kindly ask for your reconsideration of the dataset quality.
> >
> > As the deadline for the Rebuttal approaches, we want to inquire whether our responses have addressed your concerns. If any remaining problems or areas require further clarification, we would greatly appreciate it if you could provide us with additional feedback.
> >
> > Best,
> >
> > Authors of Paper 5453

---

### Author Response · Authors · 2024-12-03
**General Responses to All reviewers and AC**

Dear Reviewers and AC,

We would like to sincerely thank all the reviewers for your valuable comments and efforts. We are also happy to be acknowledged for our contributions and significance to the community. To address the concerns raised by reviewers, we have correspondingly added new experiments and case studies, which we believe have made our approach much more comprehensive and convincing.

We wish to invite all reviewers to check our new results and observations, as well as our rebuttal to support our contributions:
* **Dataset Size**: As a concentrated dataset focusing on one valuable domain, the size of our CLR-Bench, i.e., 1,018, has already satisfied the requirements to comprehensively evaluate the models' performance and make convincing and interesting observations. We have listed 6 of the former papers that are published on top-tier conferences with small number of questions/samples for your kind reference.
* **Value of CS/AI Domain**: We highly value this domain more than all the existing benchmarks for four reasons.
**First**, CS and AI naturally contain a variety of types of challenging tasks compared with all the existing 'single-type' benchmarks. In CS and AI domain, we have conceptual understanding tasks (algorithms, networks, databases, paradigms, machine learning, NLP, CV), Math problems (Discrete mathematics, linear algebra, matrix computation, calculus), Code generation problems (programming, optimization, code completion, error detection), etc. **Second**, the performance on LLMs on CS and AI is still very unsatisfactory or even bad It is valuable to benchmark and research. **Third**, it remains a blank in the benchmark of LLMs considering CS and AI. This also brings two opportunities. (i) LLMs will have few chances to be fine-tuned based on our questions. (ii) Following research on RAG or LLMs themselves could benefit a lot from our benchmark. **Fourth**, we are pioneering in a more comprehensive one in terms of multiple questions types and the difficulty in college-level. The real scenarios of applying LLMs could be much more complicated while single-type questions (in existing benchmarks) could hardly satisfy.
* **Dataset Quality**: We have multiple mechanisms to ensure the quality of dataset construction, i.e., ``Authoritative textbooks``, ``Expert-enhanced Rationale Generation`` and ``Majority Vote-based Expert Review``.
* **Evaluation Quality**: Following the widely adopted way of ``LLM-as-a-judge``[1], e.g., OlympicArena[2] and Infimm-Bench[3], we employ GPT-4 as the base referees scoring the rationale among {0,0.5,1}. **What's more**, in this paper, we propose a paradigm for more reliable expert-enhanced auto-evaluation for different types of questions. We carefully introduce the expertise to constrain the marking criteria for different types of questions through elaborate prompts and instruct GPT-4 to score the rationale. This could effectively **align** the auto-evaluation with human experts.

Thank you again for all the reviewers' suggestions that have greatly improved our paper, and we are still looking forward to receive the feedback before the deadline.

---

### Meta-Review · Area_Chair_ybF1 · 2024-12-13

**Metareview:**

This paper has several notable shortcomings. The dataset is relatively small, with only 1,018 questions, and narrowly focused on computer science and artificial intelligence, limiting its generalizability to other domains. The evaluation methodology heavily relies on GPT-4 for scoring, which introduces potential biases and lacks robustness in capturing diverse reasoning paths. Furthermore, the claimed novelty of the metrics (Q→A and Q→AR) is overstated, as similar approaches have been explored in prior works. Finally, the paper fails to provide meaningful comparisons with existing benchmarks, such as scalability issues and dataset quality assurance.

**Additional Comments On Reviewer Discussion:**

All reviewers unanimously agreed to reject the submission.

---

### Decision · Program_Chairs · 2025-01-22

Reject